# Comparison of Race Performance Characteristics for the 50 m and 100 m Freestyle among Regional-Level Male Swimmers

Łukasz Wądrzyk [1,*] , Robert Staszkiewicz [2,*] and Marek Strzała [1]

1   Department of Water Sports, Faculty of Physical Education and Sport, University of Physical Education, al. Jana Pawła II 78, 31-571 Kraków, Poland
2   Department of Biomechanics, Institute of Biomedical Sciences, University of Physical Education, 31-571 Kraków, Poland
*   Correspondence: lukasz.wadrzyk@awf.krakow.pl (Ł.W.); robert.staszkiewicz@awf.krakow.pl (R.S.); Tel.: +48-12-683-11-11 (Ł.W.); +48-12-683-15-39 (R.S.)

**Abstract:** Race analysis performed by faster and slower swimmers enables the definition of effective strategies for carrying out the competition. Until now, measurements of this type were mainly conducted among top-level athletes. The aim of the study was to determine the differences in swimming technique in sprint races between faster (FS) and slower (SS) regional-level swimmers. The performance of 33 swimmers (mean FINA points = 449) competing in 50 m and 100 m freestyle races was analysed. To determine the velocity (v), stroke rate (SR), stroke length (SL), and stroke index (SI), races were recorded with the use of cameras. Using the Student's *t*-test and Mann–Whitney U test, the results were compared for two groups: FS (mean FINA points = 557) and SS (mean FINA points = 379). In the 100 m competition, differences between groups were noticed in v (1.70 and 1.51 m/s), SL (2.06 and 1.85 m), and SI (3.52 and 2.80 $m^2$/s), while in the 50 m competition, they were noticed in v (1.95 and 1.73 m/s), SR (59.46 and 55.01 cycle/min), and SI (3.44 and 3.04 $m^2$/s, respectively for the FS and SS groups). Adapting the technique to a shorter distance should be performed by increasing the frequency of movements. At the same time, the decrease in SL should be controlled. The indicator enabling the assessment of technical effectiveness, regardless of the distance, is the SI. The 50 and 100 m freestyle races require specific technical skills to be mastered. Therefore, the development of technique in swimmers should include its various variants.

**Keywords:** swimming; biomechanics; race analysis; stroke rate; stroke length; stroke index





## 1. Introduction

Swimming races are competed at sprint distances (50–100 m), middle distances (200–400 m), and long distances (800–1500 m). Success in individual competitions is achieved only by athletes with an appropriate body build and high physical capacity corresponding to the nature of the effort (duration and intensity) [1,2]. Therefore, there are noticeable differences in these two aspects between swimmers who achieve success in different competitions (e.g., between sprinters and long-distance swimmers) [3].

It should be emphasized that realising the full bioenergetic potential is impossible without mastering proper swimming technique [4]. Characteristics of the movement course while swimming full stroke during the race are usually made by determining velocity (v), stroke length (SL), and stroke rate (SR). The mutual relations of these indicators can be described by the following equation [5]:

$$v = SL \times SR \tag{1}$$

Some authors also designate stroke index as follows [6]:

$$SI = v \times SL \tag{2}$$

the high values of which testify to the high efficiency of swimming movements [6].

It is obvious that both SL and SR are positively correlated with velocity, but their mutual correlation is negative [7]. Faster swimmers (representing a higher sports level, determined, among others, by FINA points) have been found to achieve higher speeds as a result of higher SL [8,9]. In accordance with Equations (1) and (2), faster swimmers are characterized by higher SI values. At the same time, groups representing different sports levels do not differ significantly in terms of SR [9].

The way a motor task is performed differs depending on its nature, e.g., the duration of the effort [10]. Therefore, there are noticeable differences in swimming technique depending on the distance at which the competitors race [9]. In sprint events, compared with long-distance races, competitors move at a higher speed and SR, but have smaller SI [11]. Larger SI values are usually recorded in races over shorter distances [12]. The differences described above result from the necessity to adjust movements of the upper limbs according to spatio-temporal patterns, which is visible, among others, in changing the so-called index of coordination [13,14]. The comparison of the differences in technique depending on the distance is made by collecting data on different athletes competing in other disciplines (independent samples) [8,9,12,15]. To the best of our knowledge, so far, no comparative analysis has been carried out among the same swimmers competing in races at two different distances (dependent samples). This type of approach could help verify previous reports in terms of differences in technique depending on the covered distance.

The collection of data from swimming races usually takes place during the most important national [15] or international events held in long-course (50 m) pools [5,6,16,17]. There are a number of existing works on carried out at short course (25 m) pools [15,18,19]; however, they regarded less skilled swimmers. Undertaking this type of research could, in turn, serve to formulate guidelines for swimmers who have only just started, aiming towards achieving their maximal sports results.

To date, the differences in kinematic indices for distances of various lengths have been described quite thoroughly [12,15]. They were also characterised in relation to the sports level [9]. However, it was not discussed whether swimmers representing different sports levels modify their technique in a similar way along with the extension of the covered distance. Thus, the aim of this study was to determine differences in swimming technique for sprint races between faster and slower regional-level swimmers. Therefore, the values of kinematic indices and their changes in values during 50 and 100 m races were compared. It was assumed that the technique used by faster and slower competitors will differ not only in terms of kinematic variables for the 50 and 100 m races, but also depending on the covered distance.

## 2. Materials and Methods

The performances of competitors during the 50 and 100 m freestyle races in the short-course pools were analysed. The races were filmed at a local competition held in accordance with FINA (Fédération Internationale De Natation) swimming rules. A portion of the data was obtained thanks to the Omega automatic timing system (Swiss Timing LTD, Corgémont, Switzerland). Owing to the lack of interference and the use of publicly available data for the purpose of research, it was not required to obtain consent of the athletes whose performance was analysed. It was considered sufficient to obtain approval for carrying out measurements from the main organizer—the District Swimming Association. All procedures were carried out in accordance to the Declaration of Helsinki regarding Human Research.

### 2.1. Study Group (Participants)

The inclusion criteria in the study group were the start in both events and achieving a result above 300 FINA points per 100 m freestyle [20]. Out of the total number of 384 participants of both sexes, the analysis covered film recordings of 33 male swimmers.

The results and athletes' data (name and surname, date of birth) were downloaded from the https://swimrankings.net website (accessed on 6 August 2022).

The subjects were divided into groups: those obtaining better ("faster swimmers"—FS) and worse results ("slower swimmers"—SS). The dividing line was determined by the average FINA score obtained for whole group during the 100 m freestyle race (449 points). There were 13 swimmers in the FS group (mean FINA points for 100 m freestyle = 557 and 20 in the SS group (mean FINA points for 100 m freestyle = 379).

### 2.2. Measurement Station

Before beginning the competition, points were marked on the side edges of the pool at a distance of 5, 10, 15, and 20 m from the starting wall (Figure 1). Two filming devices were placed on stable tripods: Sony DSC-RX100M3 (Sony Group, Tokyo, Japan) and GoPro Hero 9 Black (GoPro Inc., San Mateo, CA, USA). Both cameras were set at a distance of approximately 10 m from the side wall of the pool, at a height of approximately 6.5 m in relation to the water surface. Their lenses were directed perpendicularly to the direction of the subjects' movement. The Sony device (filming mode, frequency 60 frames/s, resolution 1920 × 1080) was placed in such a way as to record the course of the athletes' movements in the zone 5–10 m from the starting wall. The GoPro camera (linear filming mode, frequency 60 frames/s, resolution 1920 × 1080) was set in such a way making it possible to register the course of motion over a distance of 15–20 m from the starting wall. At the time of the starting signal, the starter judge was visible in the lenses of both devices, using the starting system emitting both a sound and a light signal to indicate the beginning of the race.

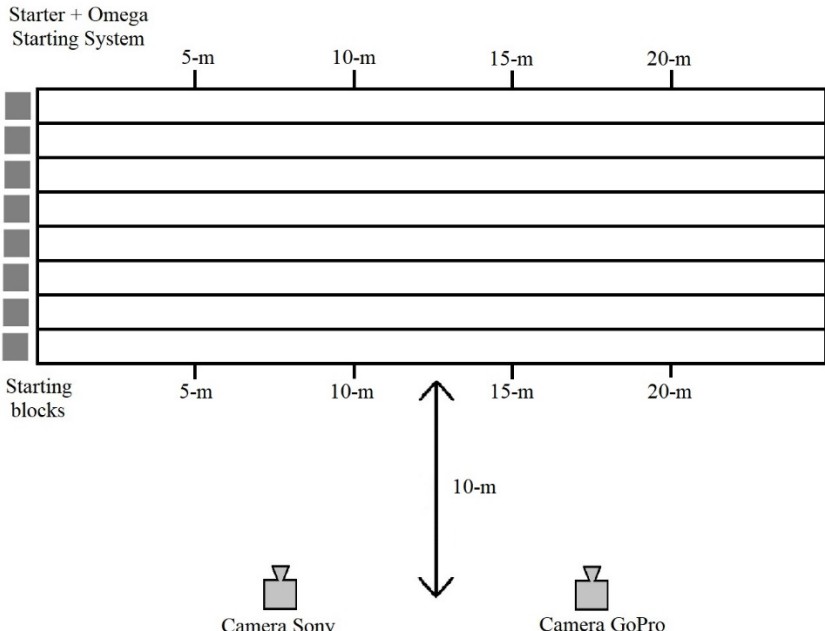

**Figure 1.** Arrangement of cameras in relation to the swimming pool.

### 2.3. Determining Kinematic Variables

The recordings were subjected to kinematic analysis using the Kinovea program (version 0.8.15, Joan Charmant & Contrib, Bordeaux, France). First, the time to cover the 5 m sections ($T_{5M}$) was determined: 15–20 m; 40–45 m; and, for 100 m races, additionally 65–70 m and 90–95 m. The sections were measured on the basis of the line connecting the markings on the side wall of the swimming pool. The time to cover these sections was estimated when the competitor's head passed the lines. The time to perform three complete movement cycles in the 5 m zone ($T_{3C}$) was also determined. In this case, the limits of one cycle were consecutive, successive submersion of the same hand in the water. Using MS

Excel (Microsoft Corporation, Redmond, WA, USA), mean values were calculated for each 5 m section:

− velocity: $v = 10/T_{5M}$ [m/s];
− stroke rate: $SR = 60 \times 3/T_{3C}$ [cycle/min];
− stroke length: $SL = v \times 3/T_{3C}$ [m];
− stroke index: $SI = v \times SL$ [m$^2$/s].

On the basis of each 5 m section, the average values of the indexes were determined for both races because they were used in these sections and not in measurements based on data from the literature:

− velocity: $v_{50}$ and $v_{100}$;
− stroke rate: $SR_{50}$ and $SR_{100}$;
− stroke length: $SL_{50}$ and $SL_{100}$;
− stroke index: $SI_{50}$ and $SI_{100}$.

The above-mentioned variables were used to determine the differences in values of indicators for both distances called Dv (differences in velocity), DSR (difference in stroke rate), DSL (difference in stroke length), and DSI (difference in stroke index):

$$Dv = v_{50} - v_{100} \tag{3}$$

$$DSR = SR_{50} - SR_{100} \tag{4}$$

$$DSL = SL_{100} - SL_{50} \tag{5}$$

$$DSI = SI_{50} - SI_{100} \tag{6}$$

where $v_{50}$—mean velocity of fullstroke swimming zones for the 50 m freestyle race, $v_{100}$—mean velocity of fullstroke swimming zones for the 100 m freestyle race, $SR_{50}$—mean stroke rate for the 50 m freestyle race, $SR_{100}$—mean stroke rate for the 100 m freestyle race, $SL_{50}$—mean stroke length for the 50 m freestyle race, $SL_{100}$—mean stroke length for the 100 m freestyle race, $SI_{50}$—mean stroke index for the 50 m freestyle race, and $SI_{100}$—mean stroke index for the 100 m freestyle race.

*2.4. Statistical Analysis*

Statistical analysis was performed via the Statistica program (version 13, StatSoft, Kraków, Poland). Differences between groups were assessed for the following variables: time to cover the 100 and 50 m freestyle races ($T_{100}$, $T_{50}$), age (A), $v_{100}$, $v_{50}$, $SR_{100}$, $SR_{50}$, $SL_{100}$, $SL_{50}$, $SI_{100}$, $SI_{50}$, Dv, DSR, DSL, and DSI. The values of all variables recorded in individual groups were tested for normality of the distribution using the Shapiro–Wilk test. In cases where the distribution of the variable was close to normal in all groups, Levene's test was applied to test the homogeneity of the variance for a given index between groups. The differences between the FS and SS groups were assessed based on the Student's *t*-test (if all the assumptions of this test were met) or the Mann–Whitney U-test (if the above-mentioned conditions were not met). Cohen's d-effect (d) was also calculated with the following thresholds: $0.01 \leq d < 0.2$—the observed effect is small, $0.2 \leq d < 0.8$—the observed effect is moderate, and $0.8 \leq d$—the observed effect is large. For all statistical procedures, a probability threshold of $p < 0.05$ was adopted [21].

**3. Results**

In Table 1, descriptive characteristics are given for the variables along with the results of the difference tests.



**Table 1.** Values of kinematic variables describing the 100 and 50 m freestyle races in the FS (faster swimmers) and SS (slower swimmers) groups, taking differences between groups into account.

| | Variable, Unit | FS (Mean ± SD) | SS (Mean ± SD) | Cohen's d-Value |
|---|---|---|---|---|
| | A * [years] | 17.09 ± 3.52 | 13.48 ± 1.10 | 1.48 |
| 100 m race | $T_{100}$ * [s] | 54.98 ± 3.21 | 62.31 ± 2.81 | 2.39 |
| | $v_{100}$ * [m/s] | 1.70 ± 0.10 | 1.51 ± 0.07 | 2.22 |
| | $SR_{100}$ [cycle/min] | 49.78 ± 2.47 | 49.21 ± 4.40 | 0.15 |
| | $SL_{100}$ * [m] | 2.06 ± 0.18 | 1.85 ± 0.16 | 1.21 |
| | $SI_{100}$ * [$m^2/s$] | 3.52 ± 0.51 | 2.80 ± 0.28 | 1.80 |
| 50 m race | $T_{50}$ * [s] | 24.91 ± 1.56 | 28.12 ± 1.33 | 2.19 |
| | $v_{50}$ * [m/s] | 1.95 ± 0.14 | 1.73 ± 0.07 | 2.06 |
| | $SR_{50}$ * [cycle/min] | 59.46 ± 4.14 | 55.01 ± 5.10 | 0.91 |
| | $SL_{50}$ [m] | 1.86 ± 0.11 | 1.83 ± 0.15 | 0.21 |
| | $SI_{50}$ * [$m^2/s$] | 3.44 ± 0.33 | 3.04 ± 0.29 | 1.27 |
| Differences in indicators for the 100 and 50 m races | Dv [m/s] | 0.25 ± 0.07 | 0.22 ± 0.03 | 0.21 |
| | DSR * [cycle/min] | 9.68 ± 4.07 | 5.80 ± 2.70 | 1.14 |
| | DSL * [m] | 0.20 ± 0.15 | 0.03 ± 0.09 | 1.40 |
| | DSI * [$m^2/s$] | 0.08 ± 0.29 | -0.24 ± 0.16 | 1.41 |

*—statistically significant difference at $p < 0.05$.

The subjects from the FS and SS groups differed in terms of age. Regarding the variables from the 100 m race, differences were found for the $T_{100}$, $v_{100}$, $SL_{100}$, and $SI_{100}$. For the above-mentioned indicators, a high value of Cohen's d-effect was noted. The $SR_{100}$ index was similar in both groups.

For the 50 m distance, the subjects differed in terms of $T_{50}$, $v_{50}$, $SR_{50}$, and $SI_{50}$ (large effect sizes). No differences were found for $SL_{50}$.

Differences between groups in terms of DSR, DSL, and DSI (large effect sizes for all mentioned variables) were noted. It was not shown that the Dv index differed significantly between the groups (moderate effect size).

Figures were constructed in which the described values are expressed in percentages (the values of variables from the FS group were assumed as 100%). These results are presented in Figures 2–4.

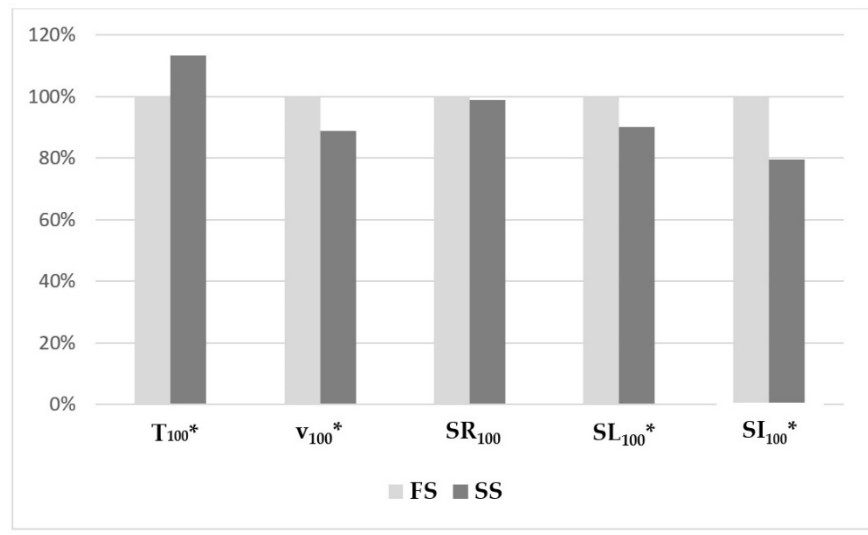

**Figure 2.** Values of indices in FS (faster swimmers) and SS (slower swimmers) groups for the 100 m freestyle race in percentages (the value for the FS group was assumed as 100%). $T_{100}$—time of 100 m race, $v_{100}$—mean velocity, $SR_{100}$—mean stroke rate, $SL_{100}$—mean stroke length, $SI_{100}$—mean stroke index. *—difference between the FS and SS groups at $p < 0.05$.

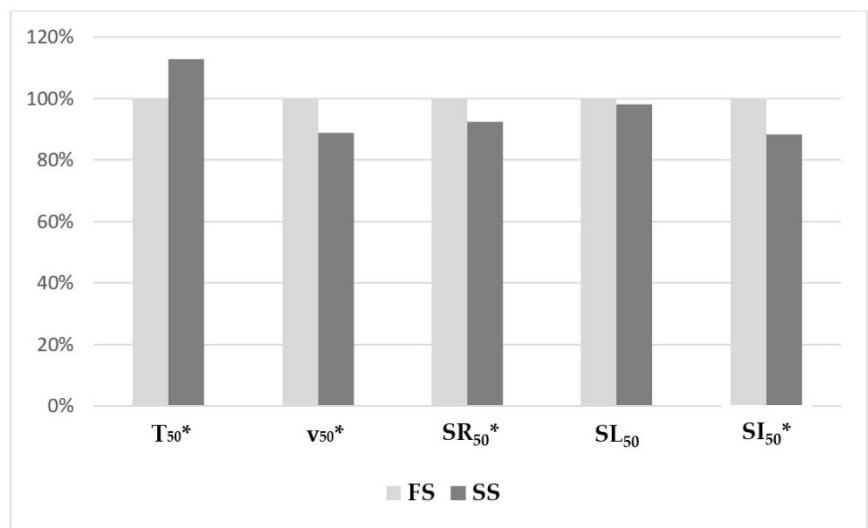

**Figure 3.** Values of indices in FS (faster swimmers) and SS (slower swimmers) groups for the 50 m freestyle race in percentages (the value for the FS group was assumed as 100%). $T_{50}$—time of 50 m race, $v_{50}$—mean velocity, $SR_{50}$—mean stroke rate, $SL_{50}$—mean stroke length, $SI_{50}$—mean stroke index. *—difference between the FS and SS groups at $p < 0.05$.

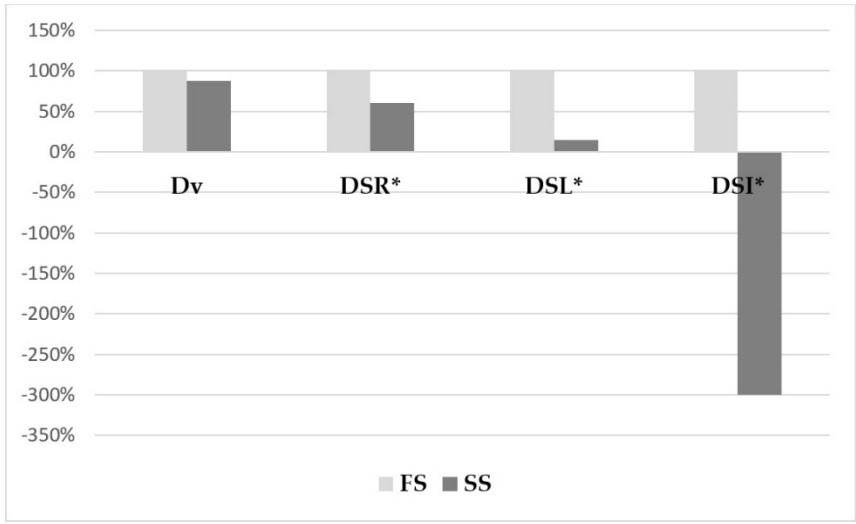

**Figure 4.** Values of differences in the FS (faster swimmers) and SS (slower swimmers) groups in indices for the 50 and 100 m freestyle races in percentages (the value for the FS group was assumed as 100%). Dv—difference in (mean) velocity, DSR—difference in (mean) stroke rate, DSL—difference in (mean) stroke length, DSI—difference in (mean) stroke index. *—difference between the FS and SS groups at $p < 0.05$.

## 4. Discussion

In this study, a comparison was made of the full style swimming technique for 50 and 100 m freestyle races performed by faster and slower regional-level swimmers. In the case of both distances, the technique of the FS and SS subjects differed in terms of v and SI. In addition, differences in SR were noted in the shorter race, and in SL, with regard to the 100 m race. It was also noted that the participants used different strategies for modifying the technique depending on the covered distance. In the FS group, the athletes modified SR and SL to a greater extent than in the SS group. Differences in adjusting SI were also noted—as in the case of the SS group, higher values were assumed over the longer distances, while in the FS group, an opposite situation occurred.

The criteria applied to select the subjects for the groups (time per 100 m) obviously resulted in the fact that, in both competitions, FS and SS differed in terms of swimming velocity ($v_{50}$ and $v_{100}$). However, it should be emphasized that, apart from the full swimming style zone, the final result is also influenced by the effectiveness of starts and turns [22,23]. Therefore, it cannot be ruled out that differences between groups may concern not only the full style swimming effectiveness, but also the start and turn zones. It is worth taking this fact into account when interpreting the results.

The age of the participants is also of relevance for the obtained results. Swimmers from the FS group were older than those from the SS group. As the complexity of the task increases (e.g., swimming at maximal or almost maximal speed), because of the small range of allowed technical imperfections, the number of possibilities to correct task performance becomes limited [24]. As pointed out by Silva et al. [25], the ideas of the "expert concept" in relation to young athletes are not appropriate, as in their case, it is often difficult to define the ideal technical solution that they should be implemented. This means that there is restricted coverage regarding the results of this study and their relevance to high-level athlete testing.

### 4.1. Differences in the 100 m Race

The higher technical level of swimmers is often associated with greater $SL_{100}$ values. Within the context of the 100 m distance, the results of this study confirmed these considerations—swimmers achieving shorter times in this race presented larger SL. A similar observation was made by Morais et al. [6] and Durović et al. [15] in groups representing higher sports levels. According to some researchers, better swimmers, by properly positioning their limbs under water, propel the body more efficiently, covering a greater distance in one cycle [26].

Another indicator used to assess the economy of movement is $SI_{100}$ [27]. This variable, alike $SL_{100}$, had higher values in the FS group. It may be the effect of effort time, which, for the distance of 100 m in the FS and SS groups, lasted about 1 min. In this type of motor task (lasting longer than 40 s), it is impossible to use the "all out" strategy, sufficiently used in shorter efforts [28]. The necessity to tolerate a high level of fatigue in the final part of a 100 m race requires rational distribution of effort throughout the duration of the task. Obtaining a better result in such a situation is possible in the case of high motor economy (low energy cost), which can be described by SL and SI [29].

There was no difference between the two groups in terms of $SR_{100}$. These results are consistent with the those obtained by the authors of previous studies for this distance [6,12]. Nevertheless, Durović et al. [15] noticed that, at the national level, slower swimmers achieved higher SR than faster competitors in the 100 m freestyle races. In swimming, the frequency of movements is sometimes equated with the intensity of exercise [30]. This means that the SR values should not be assumed as an indicator of a swimmer's technical proficiency. Nevertheless, bearing in mind the negative correlation between SL and SR [7], covering greater distances in one cycle while maintaining a comparable frequency of movements is the domain of better swimmers, which was confirmed in this study.

### 4.2. Differences in the 50 m Race

For a shorter distance, the participants from the FS group presented a significantly higher SR level than among SS athletes. In the group of 14–16-year-old swimmers, Silva et al. [25] observed differences similar in nature between the faster and slower athletes competing in the 50 m freestyle. It should be emphasized that the SS recorded low values for this indicator (approximately 55 cycles/min) during this competition. Among the best swimmers, the SR index for this distance is approximately 60 cycles/min [5,31].

With the increase in SR, in the upper limb movement cycle, athletes first reduce the duration of the non-propulsive phases (entry, stretch, and recovery) [32]. This modification results in changes in the technique of so-called negative coordination (also known as "catch-up") towards positive coordination (also referred to as "kayaking" or "superposition") [14].

The increase in SR has a positive effect on velocity, which, however, causes a significant increase in drag—especially frontal and wave resistance (increase to the square and cube of velocity, respectively) [33]. However, for such a short effort as in the case of the 50 m freestyle distance, more important than the economy of movement (low energy cost of effort) is the swimmer's ability to generate the highest mechanical power [29,34]. Therefore, a strategy based on swimming with high SR for short distances should achieve good results.

The relationship between SR and SL has been described quite well in the literature [7]. An increase in one of these indicators entails a decrease in the value of the other. This is as a result of shortening the duration of one of the propulsive phases—(push) [32]. The aforementioned increase in swimming velocity obtained by increasing the SR is thus only possible with a simultaneous, slight decrease in SL. Contrary to the results of studies by other authors [5,15], no differences in SL were noted between faster and slower swimmers. This indicates that, for the 50 m freestyle competition, strategies for achieving better athletic performance may differ depending on the athletic level of the group. Regarding regional-level athletes (according to Ruiz Navarro et al. [20] criterion, below 700 FINA points), increasing the frequency of movements may serve to improve the sports score for the 50 m distance. Nevertheless, this cannot come at the cost of a significant decline in SL.

As in the case of the 100 m distance, the $SI_{50}$ had higher values in the FS group. This is owing to the similar values of the $SL_{50}$ in both groups and significantly higher velocity in the FS group. Thus, it seems that the SI is useful in the context of assessing technical proficiency for both the 50 and 100 m freestyle distances.

### 4.3. Differences in Adapting the Technique Depending on the Distance

The Dv, DSR, DSL, and DSI indicators were proposed in order to highlight the differences in swimming technique depending on the length of the competition. So far, it has been established that competitors participating in sprint races swim at higher v, SR, SI, and lower SL values than during middle- and long-distance events [9,12]. In this study, it was also decided to investigate the differences in adjusting the technique to the distance between faster and slower swimmers. The Dv describing the size of the differences in velocity achieved for the 50 and 100 m distances in the FS and SS groups demonstrated similar values. Despite this similarity, the strategies employed to adapt the technique in the 50 and 100 m races differed. The subjects from the FS group modified their technique to a greater extent, which was reflected in the change in DSR and DSL values. In the SS group, the differences in these indicators for both distances were small. At the same time, in the FS group, the SI was higher in the 50 m than in the 100 m race, while in the group of slower swimmers, the SI values were higher for the longer distance. This may indicate that the differences in techniques between faster and slower swimmers are more pronounced during the performance of difficult tasks (swimming at or near maximal speed) than for easier efforts (e.g., moderate intensity). The differences are justified in the system-theoretical model of human motor behaviour [35].

Analysis of the aforementioned disproportions cannot be made without taking the differences in the sports level of the respondents into account. As already mentioned, for both the 50 and 100 m distances, the velocity of the subjects from the FS group was clearly higher than among the SS. This means that the FS subjects had to overcome significantly higher resistance values than the SS swimmers during the race, and this was possible owing to a significant increase in propelling force [34]. Its rise, in turn, can be achieved by increasing SR [36]. Therefore, it seems that the FS conformed to the 50 m distance requirements better than the SS.

A significant modification of the technique depending on the distance covered in the FS subjects was the smaller SL in the 50 m freestyle competition (by 0.2 m). In the SS group, the SL value at this distance was slightly lower than at 100 m. It should be emphasized, however, that in the case of the 100 m freestyle, the differences between the groups were large (about 10%), while for the shorter distance, this difference was small (below 2%). At the same time, at 50 m, the decrease in SL for the group of faster swimmers

was compensated by a significant increase in SR. It seems that a different mechanism was used by the SS, slightly increasing the SR while maintaining a more stable SL. This allowed them to increase the speed for the 50 m distance compared with 100 m by approximately 0.22 m/s (in the FS group, by 0.25 m/s). The research results clearly confirm the earlier report by Chollet et al. [13], in which it was stated that the swimmers demonstrating a higher sports level of sports dispose of a wider range of tactical and technical solutions, which are selected according to the requirements of a given competition. Moreover, in the case of better swimmers, the change in technique does not affect movement efficacy, as evidenced by the slight difference in the SI over the shorter and longer distances. Among less skilled swimmers, lower values were achieved for this variable over shorter distances, which indicates a lower ability to adapt the technique to more difficult conditions of the task (the need to achieve maximal speed).

Finally, certain limitations of this study should be emphasized. The main subject of the work was determining characteristics of the full stroke swimming technique; however, the results obtained for the 50 and 100 m freestyle were also influenced by the quality of the execution of starts and turns. Owing to some methodological limitations, the description of the technique does not include, among others, specifics of lower limbs movements or frequency of breathing. A further direction of research should be to include these aspects in research. It also seems reasonable to undertake research in which the same athletes (dependent samples) compete over longer distance (e.g., 100 and 200 m freestyle) and to carry out studies on other swimming styles or among female athletes.

## 5. Conclusions

Despite 50 and 100 m freestyle races often being classified under the common category of "sprint" races, they differ in terms of course of movement. This means that it is impossible to use a universal technique suitable for both distances. Improving the technique of sprinters competing at different distances should consider a broad range of skills. Regarding the 100 m distance among regional-level swimmers, care should be taken to increase SL. For the 50 m distance, successful swimmers are able to show higher SR. Adapting the technique to the shorter distance among regional-level athletes should thus be done by increasing the frequency of movements. At the same time, parallel to the increase in SR, the decrease in SL should be controlled. The stroke index is an indicator enabling the assessment of technical effectiveness, regardless of distance.

**Author Contributions:** Conceptualization, Ł.W.; methodology, Ł.W. and R.S.; software, Ł.W.; validation, Ł.W.; formal analysis, Ł.W., R.S. and M.S.; investigation, Ł.W., R.S. and M.S.; resources, Ł.W., R.S. and M.S.; data curation, R.S.; writing—original draft preparation, Ł.W., R.S. and M.S.; writing—review and editing, Ł.W., R.S. and M.S.; visualization, Ł.W.; supervision, Ł.W.; project administration, Ł.W.; funding acquisition, R.S. All authors have read and agreed to the published version of the manuscript.

**Funding:** Akademia Wychowania Fizycznego im. Bronisława Czecha w Krakowie (Award number(s): 022/RID/2018/19). The publication was financed within the program of the Minister of Science and Higher Education under the title "Regional Initiative of Excellence" (in Polish: "Regionalna Inicjatywa Doskonałości") within the years 2019–2022, project No. 022/RID/2018/19.

**Institutional Review Board Statement:** The study was conducted in accordance with the Declaration of Helsinki. Ethical review and approval were waived for this study owing to the lack of invasiveness of the method and obtaining the consent of the competition organizer.

**Informed Consent Statement:** Patient consent was waived owing to the lack of invasiveness of the method and obtaining the consent of the competition organizer.

**Data Availability Statement:** Not applicable.

**Conflicts of Interest:** The authors declare no conflict of interest.

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
