# Peer review of "Comparison of Race Performance Characteristics for the 50 m and 100 m Freestyle among Regional-Level Male Swimmers"

_applsci, doi:10.3390/app122412577_

Round 1

Reviewer 1 Report

Dear Authors,

 The study entitled: “Comparison of Race Performance Characteristics for the 50-m and 100-m Freestyle Among Regional-Level Male Swimmers” aims to determine the differences in swimming technique in sprint races between faster and slower regional-level swimmers. As an original investigation, it is an interesting study which concludes useful information for swimming training and competition regarding the performance characteristics of sprint swimmers.

I think the study has good ideas and an interesting approach and it provides relevant information for the subsequent performance analysis and its application for training. In my opinion, some information may be better presented. The discussion section is well organized. These concerns can be easily resolved by the authors.

Minor revisions with specific comments and suggestions are detailed below.

Abstract

Please specify the FINA points for the faster and slower swimmers as it should be different and it is the aim of the study.

Line 17: The number 2 should be written “two”. In the same line, if you define the swimmer as:

FS (Faster Swimmers) and SS (Slower Swimmers) it should be written the first time it appear in the abstract, not in the middle of it.

Line 18: “…differences were noticed..” between what ? please clarify.

Lines 19-21: I think the results could be improved.

Introduction

Line 28: Please use “.” instead of “,” in the numbers.

Line 31: The number 2 should be written “two”.

Line 35: What do you mean with “full style” ?

Lines 38,40: I think that the equations should be numbered. The first has a reference and the second no, please insert it or remove from the first and write it on the text.

Lines 44-45: Please detail the sports level.

Line 50: Stroke lengths is already define, please use SL. The same for stroke index.

Line 55: Please include some references according to the sentence “The comparison of the differences in technique depending on the distance is made by collecting data on different athletes competing in other disciplines (independent samples)”.

Line 57: The number 2 should be written “two”.

Line 61: Please change “50-metres” by 50-m as in the rest of the manuscript.

Line 63: Please include some references regarding the studies which investigated the topic in short-course.

Material and methods

Line 77: Please define FINA.

Line 83: As you referred to regional-level swimmers, I strongly recommend to include the swimming performance level according to the recent study of Ruiz-Navarro et al., 2022. It clarifies the swimming level according to FINA points.

Ruiz-Navarro, J.J., López-Belmonte, Ó., Gay, A., Cuenca-Fernandez, F. and Arellano, R., 2022. A new model of performance classification to standardize the research results in swimming. European Journal of Sport Science, pp.1-11. https://doi.org/10.1080/17461391.2022.2046174

Lines 87-88: Please include the FINA points and the swimming level classification recommended above for faster and slower swimmers.

Lines 88-89: Please rephrase, it is difficult to understand.

Line 93: Please use the same throughout the manuscript e.g., 5-,10-,15- and 20-m.

Line 96: The same here and in the Figure 1.

Lines 119-135: I think this part could be improved, in a table or writing equations with its respective number and detailing below every equation which means each acronym. Also Dv, DSR, DSL and DSI are not defined.

Lines 139-140: Please keep the order, always 100-m and then 50-m or the opposite. First you write (T100, T50), and then v50, v100.

Lines 148-150: I think these lines could be written in the text, it is not necessary hyphens.

Results

Lines 158-160: You should not repeat information, the d value is already in the table 1.

Lines 167-168: “In order to better illustrate the discrepancy in the values of indices recorded for the 167 groups” this sentence is not necessary.

Table 1.  All the acronyms should be described in the legend of the table. Also in the legend, it is not necessary to write “statistically significant”, it is implicit, significant is enough but explain between what, e.g. “difference between xxx and xxx (p < 0.05).

Figure 2. Define FS and SS in the legend of the figure.

Figure 3. Define FS and SS in the legend of the figure.

Figure 4. Define FS and SS in the legend of the figure. In general, I recommend to improve the appearance of the three figures if possible.

Discussion

Lines 185-193: I think this first paragraph should be rewrite highlighting the results related to the aim, difference observed, etc.

Lines 195-196: Numbers already in the table 1, please do not repeat information.

Line 204: Stroke length is already defined as SL.

Line 201: Please write one instead of 1.

Line 211: Stroke index is already defined as SI.

Lines 213-214: Numbers already in the table 1, maybe you can report % of difference or other.

Line 222: What means “lower level” ? Include the FINA points or the classification recommend above.

Line 228: It is written “swimmers, which was confirmed in this study” but in this study were not performed correlations, so how could you confirmed it ? It would be interesting to analyze it.

Line 231: “higher SR level” compared to what ? Please clarify.

Line 245: What do you mean with “effective” ?

Line 268: Use FS instead of faster swimmers because it is already defined.

Lines 269-270: This is interesting, you could explain more or express their usefulness for training.

Lines 283-184: Regarding the numbers, the same comment previously.

Line 294: “Among swimmers”, specify if faster or lower or less skilled swimmers.

Conclusions

Lines 298-299: I think this sentence is not adequate here because this research did not study any energetic variable, so please change it.

Line 303: Write SL where it is written stroke length. The same for the rest of the conclusion for stroke rate and stroke index. 

Author Response

Dear Reviewers,

We are very grateful for the extremely substantive assessment of the article by the Reviewers. Thanks to their efforts, the article gained significantly in value. We followed nearly all of the guidelines suggested by the reviewers. They include:

  1. Abstract:

- line number 15: explanation of abbreviations (FS and SS) has been added

- line number 18: the number “2” replaced “two” (the same in the: lane number 35, 64, 246

- line numbers 19-20: added mean FINA points for the FS and SS groups

- line number 20: added “between groups” to clarify the differences

- line numbers 21-22: added mean values of indicators

  1. Introduction:

- line number 38: clarified the term "full style" as "full stroke"

- line numbers 42-45: changed the formatting of equations (similarly in line numbers 150-154)

- line numbers 48-49: added a way to determine sports level

- line numbers 50-51: described relationship of the equations 1 and 2 and its application

- line number 56: full names of variables have been replaced with abbreviations (the same in line numbers 240, 247, 359, 360, 362 and 363)

- line numbers 62-63: added some references

- line number 68: changed “50 m” for “50-m” (the same was done in the rest of the manuscript)

- line numbers 69-71: references were added, slightly changing the order of the sentence

- line numbers 82-84: a slightly more detailed description of the assumed purposes of the study has been added

  1. Materials and methods:

- line number 88: added full name of FINA

- line numbers 90-93: research ethical issues were described in more detail

- line number 98: added reference of Ruiz-Navarro et al. (2002) work

- line numbers 102-105: split one sentence into two to make it clearer

- line numbers 105-106: included mean FINA points for the FS and SS groups

- line numbers 141-143: added “because they were used in these sections and not in measurements based on data from literature”

- line numbers 155-160: a description of the variables has been added under the equations

- line numbers 164-165: the order of the variables has been changed

- line numbers 173-175: hyphens were deleted

  1. Results:

- line numbers 182, 183, 185, 186, 188: d-value were rejected,

- line numbers 192-193: unnecessary beginning of sentence has been removed

  1. Discussion:

- line numbers 214-221: a fragment describing the most important research results has been added

- line numbers 229-231: in order to remove duplication of information from the "Results" section, the numbers have been removed (the same in line numbers: 249, 329-330)

- line numbers 258-259: the sports level was clarified in the cited studies

- line number 263: for a better description of the phenomenon, the publication of Craig et al was quoted

- line number 268: a reference group has been defined

- line number 272: updated bibliographic references to more recent ones (the same in line numbers: 277, 281, 316)

- line numbers 282-283: the term “effective” was clarified

- line numbers 300-305: the indicators used in the work were referenced to bibliographic sources

- line numbers 313-316: an attempt was made to indicate the reasons for the differences between the FS and SS groups in terms of the SI index

- line numbers 344-352: partially corrected in accordance with review 3, adding limitations of the study and future perspectives. However, this was not done in the form of a separate chapter due to the editorial guidelines regarding the preparation of the manuscript

  1. Conclusions:

- line number 355 - removed the term "bioenergetic requirements"

  1. References:

- this section has been corrected in accordance with the changes throughout the text

After careful consideration, it was decided not to change:

- figures 2-4 (line numbers 195-211) - unfortunately, it is not possible for us to improve the quality of the figures due to the need to use atypical abbreviations with subscripts. We realize that they may raise objections. We hope, however, that the current quality of the figures engravings is satisfactory,

- line numbers 249-250: the first review suggested: “Lines 213-214: Numbers already in the table 1, maybe you can report % of difference or other” - we would like to point out that in this case, reference to between-group percentage differences should not be made, as we are describing a phenomenon that applies to both groups (effort over 40 s),

- the sentence in line numbers 360-362 – the second review suggested: “In paragraph 304, of concussions, cerebrovascular accidents are mentioned, but this cannot be verified in their study” - we consider this a possible error when creating a review (perhaps the sentence was taken from a review of another article?).

As required, all corrections were made in the change tracking mode. We sincerely hope that the current version of the article is fully compliant with the publication requirements of Applied Sciences.

Yours faithfully, on behalf of the authors,

The main author

Reviewer 2 Report

In paragraphs 38 to 40, describe the relationship, if any, of the equations, and be more explicit in its application.

In paragraph 124, because it was used in sections of 5 meters and not in other measurements based on the literature.

In the discussions, relate the findings based on the literature based on the indicators used.

In paragraph 304, of concussions, cerebrovascular accidents are mentioned, but this cannot be verified in their study.

Author Response

(The authors gave the same response as above.)

Reviewer 3 Report

First of all, I would like to thank you for the opportunity to review this very interesting article. I certainly believe that its publication is necessary as it is of a high scientific standard, but I think that it needs some minor modifications.

The introductory section is very well written, but I think that the research objectives and hypotheses should be stated at the end of this section. 

In the material and methods section, I recommend adding a new section entitled "procedure" where the context of how the participants were contacted is described. I also recommend adding the ethical aspects of the research. 

Regarding the presentation of the results, these have been carried out in a clear and concise manner. Focusing attention on the discussion, I recommend adding a more current quote from research carried out in the last 5 years. 

Likewise, I also recommend adding a section entitled limitations and future perspectives, where some of the limitations of this study are pointed out and which this research adds to the proposed topic. 

Author Response

(The authors gave the same response as above.)
